# Collective Action for the Market-Based Reform of Land Element in China: The Role of Trust

Lin Zhou * and Walter Timo de Vries

Chair of Land Management, School of Engineering and Design, Technical University of Munich (TUM), 80333 Munich, Germany; wt.de-vries@tum.de
* Correspondence: lin.zhou@tum.de

**Abstract:** The market entry of collectively-owned operating construction land (COCL) is an important policy of the Chinese government to promote the flow of rural land elements in the market. Describ-ing, characterizing, and understanding collective action for COCL marketization in China is conducive to identifying potential contradictions in a timely manner, constructing common goals, and promoting stakeholder cooperation to improve the efficiency of land marketization. Our research question is to identify which conceptual and theoretical models would be most appropriate to evaluate the market-based land reform in China. Relying on a narrative review approach, we interpret the literature and infer that trust is conducive to cracking the collective action puzzle of COCL marketization, and propose a conceptual or theoretical framework for the joint analysis of social capital, trust, and cooperation performance for modeling and investigating the important role of trust in collective action. Concentrating on the role of social rationality in land marketization, we suggest a pathway to break away from the collective action dilemma focusing on land property rights to build stakeholder trust relationships. Subsequent research could continue by developing indicators to measure social capital, trust, and cooperation performance and empirically investigate the relationship between them on this basis.

**Keywords:** collective action; land marketization; collectively owned land; trust; theoretical framework

## 1. Introduction

Sustainable development has long been an important global issue. In 2015, the United Nations launched the 2030 Agenda for Sustainable Development, with 17 global sustainable development goals (SDGs). Land use and distribution have an impact on the environment, making the topic of effective land use management in the light of sustainability particularly important. In China, the government has enacted many land policies to improve farmers' lives but has often encountered obstacles in promoting the implementation of the new land policy. With regard to the formulation and implementation of land policies, villagers are almost exclusively concerned with how much they can gain and therefore rarely express their true views on existing land policies, although many policies and literature mention that the implementation of land policies should respect the wishes of villagers and advocate public participation.

At the same time, one can observe a decline in China's economic growth rate. There is a smaller labour force and an increasing reliance on the surplus rural labour, which leads to a lower savings rate and an aging population in the rural regions [1–3]. To this end, the government applies an approach to shift the economic development from factor input driven to productivity driven [4]. In addition, it promotes innovation by stimulating market-oriented reallocation of production factors. Among the main factors of production, collective land is generally not allowed to be traded in the land market due to strict government control and can only be traded in the land market after land expropriation, which changes collective land ownership to state-owned land ownership [5]. This land

acquisition policy has facilitated the development of urbanization and industrialization, but it has also resulted in the de-agriculturalization of agricultural land, excessive expansion of urban scale [6,7], a wide gap between urban and rural development, and damage to the interests of landless farmers [8,9]. In particular, as a large number of farmers have moved to the cities, the shortage of land in urban areas has led to soaring real estate prices, while the phenomenon of idle and abandoned land in rural areas has become increasingly serious [10].

China's underdeveloped rural land market is unable to foster an effective mechanism of supply and demand. This has prompted the Chinese government to open up the rural land market and carry out market-oriented reforms of land factors by proposing a series of policies and regulations [11,12]. In 2013, the establishment of a unified urban-rural construction land market was proposed, allowing collectively-owned operating construction land (COCL) to enter the land transaction market, subject to planning and use control. In 2015, 33 administrative regions were selected as pilot sites for the COCL marketization reform. In 2017, the report of the 19th Party Congress proposed that the reform of the economic system had to improve the property rights system and the market-oriented allocation. On 1 January 2020, the Land Management Law broke down the legal barriers to COCL marketization. In April 2020, the Opinions of the CPC Central Committee and the State Council on Improving the Systems and Mechanisms for Market-based Allocation of Factors of Production', the first document of the Central Government on the market-based allocation of factors, gives specific guidance on market-based allocation of factors such as land, capital, labour, technology and data. In 2021, the Action Plan for Building a High Standard Market System gives prominence to the promotion of market-based allocation of land factors. The market entry of collectively-owned operating construction land (COCL) is an initiative which aims at establishing a unified construction land market for urban and rural areas (the Decision of the CPC Central Committee on Several Major Issues Concerning Comprehensively Deepening the Reform). According to statistics released at the seventh meeting of the Standing Committee of the 13th National People's Congress on 23 December 2018, as of today, in China's 33 pilot counties (cities and districts), a total of more than 10,000 pieces of COCL have entered the market, covering an area of more than 90,000 mu, with a total price of approximately RMB 25.7 billion and a reconciliation fund of RMB 2.86 billion, while a total of 228 pieces of COCL have been processed for mortgage loans, totaling RMB 3.86 billion. The incomplete and ambiguous property rights prevalent in rural China have led to peculiar land revenue distribution outcomes [13]. While farmers' perceived land tenure rights are low, due to forced evictions and government intervention, land tenure systems have significant social support and low levels of conflict. The reasons are that institutional credibility and interpersonal trust play an important role in safeguarding perceived land tenure security, allowing collective action to proceed smoothly. The COCL marketization in China is an important land policy proposed by the government to promote the marketization of rural land elements and improve the urban-rural dichotomy. Hence, describing, characterizing and understanding collective action for COCL marketization in China is conducive as it can identify potential contradictions in a timely manner, construct common goals and promote stakeholder cooperation to improve the efficiency of land marketization.

The research question of this paper is to identify which conceptual model would be most appropriate to evaluate the market-based land reform of land in China. This study draws on existing literature dealing with the dilemmas, causes and solutions to collective action in COCL marketization and extends this by constructing a conceptual and theoretical framework for collective action with trust as an intermediate variable. This framework provides the foundation for subsequent empirical analyses of the relationships between social capital, trust and cooperation performance in collective action. The objective of this inferential review is to derive an approach to solve the collective action dilemma of land marketization and to construct and understand the relationship between social capital, trust

and cooperation performance. With this, it should be possible to measure this relationship, support land marketization, and avoid collective action dilemmas.

The paper is organized as follows. Section 2 reviews the theoretical perspectives on the collective action of land marketization in China. Section 3 introduces the research area and data sources. Section 4 synthesizes findings on the identified challenges of collective action to develop a theoretical framework. Section 5 concludes by synthesizing how to address the collective action dilemma of land marketization in China.

## 2. Collective Action of COCL Marketization in China

Zhou et al. (2020) argue that in terms of interest patterns, China's current land allocation is generally at the expense of farmers' interests, and the market is not functioning as it should [5]. The Chinese government has attempted to implement COCL marketization by establishing a unified urban-rural construction land market and improving the spatial mismatch and underdevelopment of rural land factor markets. Market participants reduce the uncertainty of market formation by setting rules to accomplish activities such as process review, land transactions and revenue distribution. It is difficult for any one individual to have sufficient capacity and resources to carry out these activities, which to varying degrees require collective action. While existing studies address operational issues such as mode choice for market entry [14], distribution of benefits [15,16], and allocation efficiency [17] in COCL marketization, few provide a theoretical description and dissection of how, when and where collective action takes place in this context. In light of this [18,19], there is a need to view the process and dynamics of the COCL marketization from the perspective of collective action [20,21].

### 2.1. Collective Action Dilemmas in the Land Marketization

There are three factors that may give rise to collective action based on the General Theory of Conceptual systems [22]: Subjects, objects and environment. "Subjects" are individuals or groups involved in collective action, including landowners, land tenure holders and policy implementers who are often referred to as stakeholders in the context of COCL marketization [5,17,23]. "Object" refers to the social activity in which the subject participates, and they can all be triggers for collective action [24]. The object of COCL marketization is to the full life cycle, including pre-market entry preparation to determine land conformity and title registration; qualification review for application, review and democratic resolution; public trading; and distribution of proceeds. "Environment" means the behavior and issues that arise from the interaction between subject and object [25,26]. For COCL marketization, the environment is a mutual benefit or conflicting actions of stakeholders throughout the life cycle.

Whilst social collective action for land marketization is important, it is often difficult to achieve [27]. This is because participants exhibit speculative behaviour, operate under bounded rationality, have to deal constantly with uncertainty, and work in information asymmetries. Such conditions tend to generate transaction costs that hinder or stall collective action. One can specify the collective action dilemma in China's land marketization and the reasons for it through a number of characteristics.

The first aspect is rent-seeking behaviour. In the process of marketizing rural collective land, landowners or stakeholders tend to seek rents above the market price, which in turn leads to a reduction in the marketization of the land as it increases the transaction costs and rent negotiation times. Another manifestation of rent-seeking behaviour is that due to government restrictions on land property rights, landowners or stakeholders are unable to change the use of the land to gain additional revenues. As a consequence, they tend to circumvent the law or use the land illegally, thus increasing the workload of government intervention, investigation and problem-solving to achieve a compliant, reasonable and legal market outcome [28].

The second characteristic is the prisoner's dilemma. Since the land tenure entering the market is collective ownership, individual members of the village collective need

to negotiate to reach a consensus on willingness to cooperate. They tend to make the choices to optimize their individual interests rather than seeking a collective benefit. The dilemma with collective action is that individual members cannot overcome and restrain their selfishness to make the collective best choice for mutual benefit [29].

The third issue is free riding. Collective action cannot exclude those who do not contribute from benefiting from its development [18,30]. Individual rationality often has a tendency to free ride in the achievement of collective goals, and therefore individual rationality is not a sufficient condition for achieving collective rationality [18]. Collectively-owned land is characterized by non-adversarial and low exclusivity, and an actor's contribution to collective land has positive spillover effects that increase the overall benefit, with the benefits realized often spilling over to others. Individuals will not cooperate and thus lead to collective action dilemmas when they have the expectation that others will contribute or when they believe that their non-cooperation will not affect the supply of collective land.

The fourth assumption is low perceived returns [27]. Perceived returns refer to the impact of their contribution to the entry to the market as perceived by participants. Early contributions to land marketization often do not result in tangible benefits, and therefore participants' perceived returns to their contributions are low. As land markets gradually develop and improve, more contributions accumulate and the process and effects of land marketization only become apparent. Thus, with uncertainty and low perceived returns, the land marketization process is often hampered in the early years.

These problems derive from the behaviour of self-interest of the participants. Self-interested people only pursue their own personal benefits and do not consider the impact of their actions on the collective and society [31,32]. The Land Administration Law of the People's Republic of China (2019 Amendment) provides that COCL that complies with the plan and is legally registered requires the consent of at least two-thirds of the members or village representatives of the collective economic organization if it is to be sold and leased. If a stock of social capital—in the form of shared consciousness, mutual trust and normative agreement—cannot be accumulated among collective members, there are high transaction costs. Those who are self-interested and act in their own interest will likely externalize the costs to others, and the collective action of COCL marketization generates high transaction costs under government regulation and market mechanisms, to the detriment of the construction of a unified urban-rural construction land market. Social rationality can break through the rational economic man assumption of mainstream economics and emphasize the pursuit of individual interests along the path of collective maximization. Social rationality is both an idea that promotes 'human growth' [33] and a mode of decision-making that reconciles altruism and self-interest. It is seen as an extension of self-interest rationality [34], allowing participants in land markets to focus not only on their own interests but also to strengthen their concern for the groups and societies in which they live.

## 2.2. Trust as a Factor in Solving Collective Action Dilemmas

The execution of collective action depends on adequate information obtained through exchange within the organization. If stakeholders are characterized by rent-seeking behaviour, prisoner's dilemma, free-riding and low perceived returns as described above, significant transaction costs will be incurred in the exchange process, thus discouraging collective action [35,36].

Clear property rights to land would seem to provide answers to the above questions both at the theoretical level and in empirical studies. At the theoretical level, the new institutional economics, particularly the property rights school, sees property rights as formal rules governing people's social interactions [37,38], which do not only specify who has access to which resources under what conditions [39], allowing people to trade in a secure environment, but also provide incentives for property owners to weigh the pros and cons and use resources wisely. At a practical level, Ho (2005) notes that the lack of complete and clearly defined formal rules for rural land property rights in contemporary China

has hindered the marketization of land leases [40]. Such insecure property rights leave original property owners without the security of formal institutions [17]. Additionally, many informal or oral agreements may emerge, i.e., relational transactions based on trusting relatives and close partners [41,42]. Luo (2018) notes further that farmers' expectations of stability in land tenure are significantly reduced with trust becoming a key complement to formal institutions in the development of land rental markets [43] and an important safeguard for rural land tenure security [14].

Trust has a certain economic value in that it eliminates excessive contracting and gaming, reduces coordination costs, reduces transaction costs and increases efficiency. It also makes stakeholders willing to share information and promotes collective action and cooperation [44]. The essence of trust is the act of needing the help of others to accomplish certain events under conditions of incomplete information or limited rationality. It is the mechanism by which social rationality is formed, implying a shift from self-interested rationality to social rationality by abandoning the individual's claim to maximize self-interest. A shift from a focus on land property rights to a focus on trust is essential in order to escape the dilemma of collective action generated by self-interested behaviour. Emphasis is placed on the important role of trust in collective action, which is an important factor in villagers' support for the marketization of collective land. While existing research is beginning to emphasize institutional trustworthiness [40] and interpersonal trust [14,45], there is still a need to construct a theoretical framework for collective action of land marketization with a trust perspective.

## 3. Materials and Methods

### 3.1. Research Design

This study uses a narrative review approach [46], which allows for a broad search across different disciplines. The study analyses collective action in land marketization in China at a theoretical and literature review level and covers four areas of knowledge: land administration, land sociology, agricultural sociology, and social psychology. This approach provides a broad perspective on the study and expands its interdisciplinary scope. The data collected are secondary data obtained through literature searches of Science Citation Index database, mainly Web of Science, with "all databases" selected in the list of databases, with the aid of Google Scholar for literature searches.

### 3.2. Setting of Keywords and Terms

The literature search used the following keywords: land marketization, social capital, trust, cooperative performance, collective action, and rationality. Terms to the query preview relate to the combined relationship of different keywords, divided into three categories: limited to the two terms "land" and "market"; limited to the term "land"; and not limited to the term "land". The search terms and combinations used to find relevant literature are listed in Appendix A, and the number of searches, limited to the two terms "land" and "market", is significantly lower than the other two categories.

### 3.3. Selection Criteria

In conjunction with the two questions "what are the problems faced in the market-based reform of land element in China" and "what is the thinking for solving the problems of land marketization" addressed in Section 2, the focus was on constructing a conceptual and theoretical framework on collective action for land marketization. The research synthesis focuses on a number of logical positions in order to draw out the similarities and differences between these perspectives. With this objective in sight, there is still a need to filter out the valuable literature from the results of Appendix A. Literature was first removed on topics not relevant to this study by using Citavi during the importation of citation text files, such as online learning, intellectual property, corporate governance, renting, self-employment, trust funds, public health, history, politics, communications, energy, automation, aerospace, vehicles, dynamics, logistics, batteries, human-computer interaction,

signals, medicine, health, political economy, real estate economics, social media, business, fisheries, animal husbandry, and wildlife management, etc. Secondly, expert-recommended literature has been included to enhance the grasp of the research area. Finally, the titles, keywords, and abstracts of all selected literature were derived and subjected to a final round of screening, followed by a full review of the remaining literature. It is worth noting that the categories of social capital and trust are extracted separately in this process, and access to the concepts of the different categories facilitates an in-depth dissection of the connection between social capital and trust.

## 4. A Conceptual and Theoretical Framework for the Collective Action of Rural Land Marketization

### 4.1. Three Levels of Understanding the Collective Action of Land Marketization

This study examines the elements of collective action that influence the marketization of land at three levels—social capital, trust and cooperation performance. Our findings reveal their roles in collective action and how they can facilitate and hinder collective action in the process of land marketization.

### 4.1.1. Social Capital and Collective Action

Social capital refers to actors' relationships based on mutual trust and reciprocity as a means of reducing market, cognitive and resource uncertainty [47], generating resource flows [48] and facilitating social organization to cooperate for social efficiency [49]. Social capital is based on network relations and contains three types [50,51]: structural, cognitive and relational [52,53].

(1)　Structural dimension of Social Capital: Social network

The structural dimension refers to the social network connections between actors. Social networks are also a form of social capital, which can be seen as a social resource that exists in long-term stable network relationships and is collectively owned by members [44]. Social capital generates trust and reciprocity between individual actors through close social network interactions, resulting in cooperation in a way that breaks through the completely rational selfishness of individuals. This process becomes an important part of social capital's ability to overcome collective dilemmas. The high degree of closeness of social networks indicates frequent contact between members, which facilitates the rapid flow of information through the network and makes it easier to escape from the selfish decision-making style of members for the purpose of information exchange and collective action. The structural dimension of social capital (Figure 1) affects the actor's ability to access information and engage in action and is studied with villagers, with analysis including network position and network structure [53]. The actor's position in the network determines whether he can receive information and influences the transmission of information [54]; network structure is used to dissect the strength of the small groups in the information transfer process and the connections between them.

(2)　The cognitive dimension of Social Capital: Institutions and norms

The cognitive dimension refers to the shared goals and values of actors. Institutions and norms (Figure 2) are enforceable regulations used by groups in many forms of organizations and are specific prescriptions for collective action to regulate order and sanction behaviour that undermines rules [29]. Effective norms make it easier for people to act in the collective interest by appropriately forgoing self-interest, helping to develop interpersonal trust and a sense of community [55]. Although actors may have different strategic goals, the difference between individual goals and overall goals can be addressed by developing common goals to facilitate the effective functioning of the network. When the number of network actors increases and transaction and communication costs rise, it is easier to maintain collaborative relationships when actors share common strategic goals, values, and culture [56].

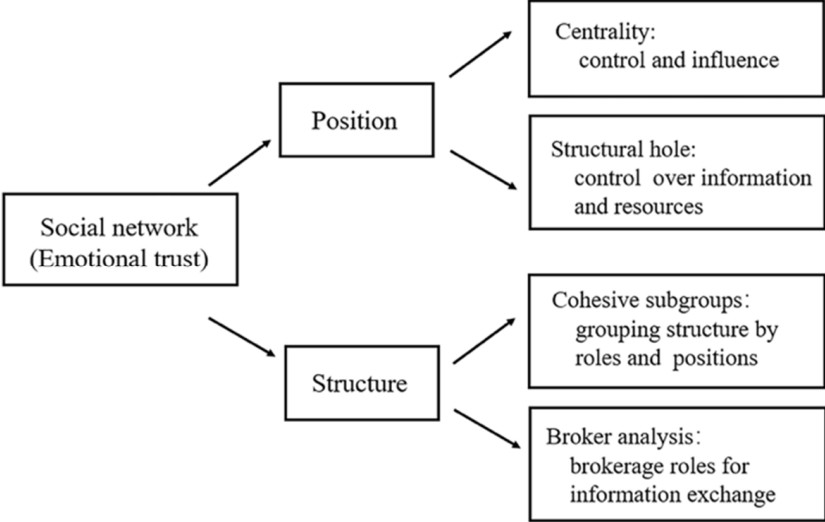

**Figure 1.** The structural dimension of social capital.

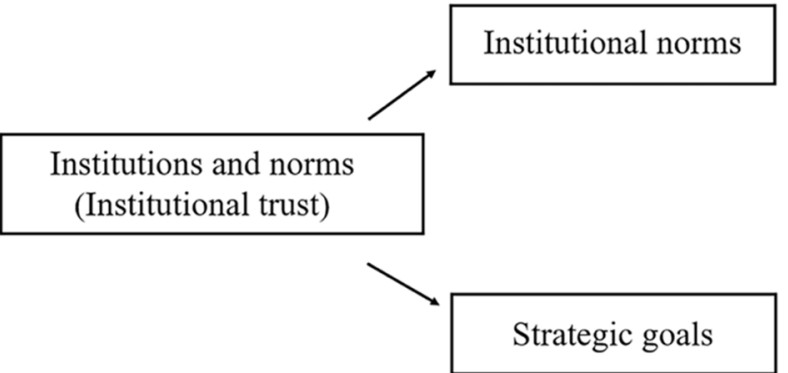

**Figure 2.** The cognitive dimension of social capital.

(3)  The relational dimension of Social Capital: Trustworthiness

The relational dimension refers to the trust between actors. Trust is not only one of the forms of social capital, but also a consequence of it, and furthermore an important factor between social capital and successful collective action [57]. Trust links social capital to collective action and is a key factor in resolving collective action dilemmas. The trustworthiness that trustees have often derived from the trustor's own ego traits and are an expression of individuality in collective action [58]. Trustworthiness is trust in the intrinsic motivations of others and is the key to trust. Trustworthiness is an important abstraction that precedes trust and can be characterized by reputation, capability, benevolence, and integrity [59]. In the initial stages, by virtue of the trustworthiness of the policy implementers, the landowners built up a one-way trust in them and then move to two-way reciprocity through close interaction and communication. Finally, landowners and policy implementers successfully overcome the collective action dilemma to culminate in a two-way cooperative relationship of mutual trust and reciprocity (Figure 3).

4.1.2. Trust and Collective Action

The dilemma of collective action lies in the potential conflict between the individual and collective interests faced by each member of a group with a common basis of interest [18]. If an individual member is not deprived of the right to enjoy the collective good, then he will have strong incentives to avoid taking responsibility for it. The collective action dilemma points to the difficulty and fragility of human cooperation, which manifests itself when there is a conflict between individual and collective interests. The solution to the

collective action dilemma requires individual members to overcome selfishness in order to achieve mutual benefits [29]. Trust eliminates excessive contracting and gaming, reduces the monetary and time costs of transactions, and allows for an effective connection between the individual and the collective to get out of the collective action dilemma [44]. Trust here refers to the likelihood that the trustor is willing to be harmed by the trustee's actions, and this willingness is based on the trustor's prediction that the trustee's actions are important to him or her, regardless of whether the trustor has the ability to control or monitor the trustee—that is, the need for trust arises in risky situations [59]. It can come from the institutional norms of society, from the social identity of the group, or from personal factors. Drawing on definitions of types of trust from various fields and schools, this thesis divides trust in the context of land marketization into three types: emotion-based, institution-based and cognition-based trust.

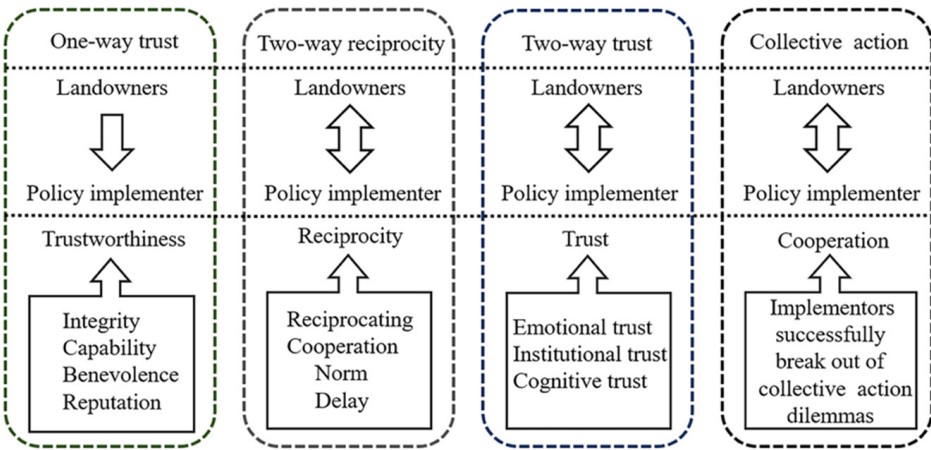

**Figure 3.** The relational dimension of social capital.

(1) Cognition-based trust

Initial expectations of the prospective partner and assessment of trust risk are prerequisites for establishing cognition-based trust. The former includes personal characteristics (e.g., gender, voice, and appearance), cultural background, behavioural motivation, professional competence, and reputation of third parties [60]; the latter is to predict the benefits and costs of cooperation [61]. The amount of trustworthiness evidence available to the trustor determines whether the trustee is trustworthy or not [62]. The trustor's comprehensive knowledge of the trustee is useful in predicting the behaviour of the trustee and making sound judgments about cooperation [63]. The cognition-based trust of land marketization is the different attitudes expressed by villagers towards the market-friendly behaviour and decisions of policy implementers through their all-around evaluation and assessment.

(2) Emotion-based trust

Emotion-based trust is formed by the emotional attachment that results from the entry of emotional factors into the relationship between the individual and the object of trust during frequent interactions over time [61]. Its main characteristic is that it is possible to develop it only after a long period of interaction. As the interaction grows closer, qualities such as goodwill and integrity come to the fore, the relationship develops steadily, and shared values are established, all of which may facilitate mutual recognition to optimize the effectiveness of the exchange between the two parties [63]. Once an emotional connection is made, the relationship of trust between them becomes stronger and there is potential for further cooperation. Emotion-based trust in land marketization is more likely to trust in relatives who also participate or vote on whether COCL enters the market and trust in villagers from the same or different villages.

(3)    Institution-based trust

"Institution-based trust" means that one believes impersonal structures support one's likelihood of success in a given situation [64]. It can develop without relying on personal traits or past records, replacing reliance on specific objects and specific exchange processes [65]. Institutional rules and ethical norms in society are the basis for the formation of institution-based trust [66], which allows expectations of future cooperation to be based on more objective criteria and the objects of exchange to become more universal [65]. The object of trust has also shifted from individuals or groups to formal institutions (e.g., legal regulations or professional certifications) or informal institutions (e.g., social or corporate culture) [67]. The institution-based trust in land marketization stems from the low level of conflict perceived by villagers due to their recognition of the institutional function of the COCL and the prediction that it will facilitate the equitable and orderly flow of rural land resources in the future.

### 4.1.3. Cooperation Performance and Collective Action

In a collective action, individual behaviour has externalities for others, in that individual and social optimality are often incompatible. When group rationality contradicts individual rationality, self-interested behaviour can prevent cooperation [18] and the imposition of negative externalities by individuals on other group members often leads to the tragedy of the commons [19], resulting in the failure of collective action. Successful collective action therefore depends on people maximizing their common interests and avoiding maximizing their individual interests. We argue that successful collective action and cooperation are equivalent concepts [68], but neither is the ultimate goal. We support Bain's reference in industrial organization to cooperation (or successful collective action) as a conduct theory [69] and believe that cooperation performance is the outcome, as it provides a criterion for judging the quality of collective action. Cooperation between actors enables collective action to achieve cooperative goals, and a good cooperative relationship brings performance to the actors. Cooperation performance takes two main forms: economic performance and social performance. Economic performance is measured quantitatively in terms of the efficiency of land resource allocation and investment, which is achieved through improved market mechanisms. Social performance is a subjective perception measure of cooperative performance and consists of three components [68]. The first is sustaining satisfaction in working partnerships, which refers to villagers' satisfaction with land productivity, profitability and the overall performance of policy implementers. The second is coordinating efforts in working partnerships, which refers to the extent to which predetermined goals, milestones, and final goals are achieved. Finally, dependence and working partnerships refer to the level of loyalty of partners and willingness to continue to participate in other partnerships.

### 4.2. The Connection between Social Capital and Trust

Trust and social capital belong to a two-sided relationship. Trust is the external expression of social capital, and social capital provides the trust with the influencing factors for social order and collective action. As shown in Table 1, the three dimensions of trustworthiness, social network, institutions and norms in social capital have become inseparable from the three types of trust.

### 4.3. Connecting Trust to the Identified Challenges to Develop a Theoretical Framework

The overall theoretical framework is based on the framework of second-generation theories of collective action proposed by Ostrom and Ahn (2003) [44] shown in Figure 4. Ostrom and Ahn (2003) [44] identify trustworthiness, networks, and institutions as three basic forms of social capital and incorporate them into a theoretical framework of collective action, proposing Second- Generation theories of collective action (Figure 4). It views social capital as existing in the form of intangible resources in the social relationships between people, which draw on beliefs such as trust, norms, and participation to accomplish the

goals of that social relationship jointly. Social capital refers to the elements within social organizations that enable cooperation to enhance social efficiencies, such as trust, norms and networks, the coordination of which is facilitated by all three to enhance social efficiency.

**Table 1.** Correlation table between social capital and trust.

| Three Dimensions of Social Capital | | Three Types of Trust | |
| --- | --- | --- | --- |
| Trustworthiness | To assess the trustworthiness of the trustee's commitment or behaviour by combining objective and subjective information such as past deeds, experience, knowledge, and trustee's personal qualities. | Cognition-based trust | The willingness to trust the other party through the perception of the trustee and the measurement of risk assessment. |
| Social network | Social capital as a social resource is embedded in a long-term stable social network relationship. Dense social networks provide the impetus for transformation between different social capitals, allowing the actor to cooperate with each other for mutual benefit. | Emotion-based trust | A relationship of mutual trust and dependence is achieved through frequent interaction over time. |
| Institutions and norms | Effective social institutions or norms limit actors' individual interests and behaviours that are detrimental to collective action and develop relationships of trust and a sense of community. | Institution-based trust | The institutional regulations and moral codes in society give actors a certain level of security. |

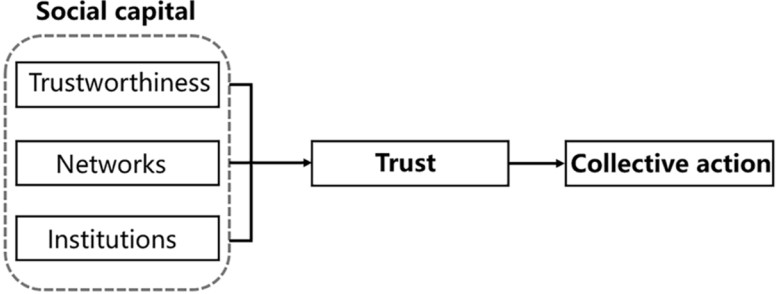

**Figure 4.** The framework of second-generation theories of collective action (Reprinted with permission from Ref. [44], 2003, Ostrom and Ahn).

Current research has relevant theoretical tenets about social capital, trust, reciprocity, collective action, and collaborative performance, but none of it adequately captures the relationship between social capital, trust, and collaborative performance. The overall theoretical framework of this thesis combines types of social capital and trust theory to reconstruct the theoretical framework of second-generation collective action and focuses on the behavioural outcomes of villagers through cooperation performance, as shown in Figure 5. The trustworthiness of policy implementers, social networks, institutions, and norms together constitute the types of social capital, representing the relational, structural and cognitive dimensions of social capital, respectively, which influence villagers' perception of social capital and choices of land marketization behaviour. The trustworthiness of policy implementers, close social networks, and proper perceptions of institutions and norms will result in villagers' land marketization strategies, in particular, whether or not to opt for trust. Good trust relationships motivate villagers as landowners to cooperate with policy implementers and land tenure holders in land transactions, and cooperative performance is a criterion for evaluating the outcomes of villagers' land-marketing behaviour.

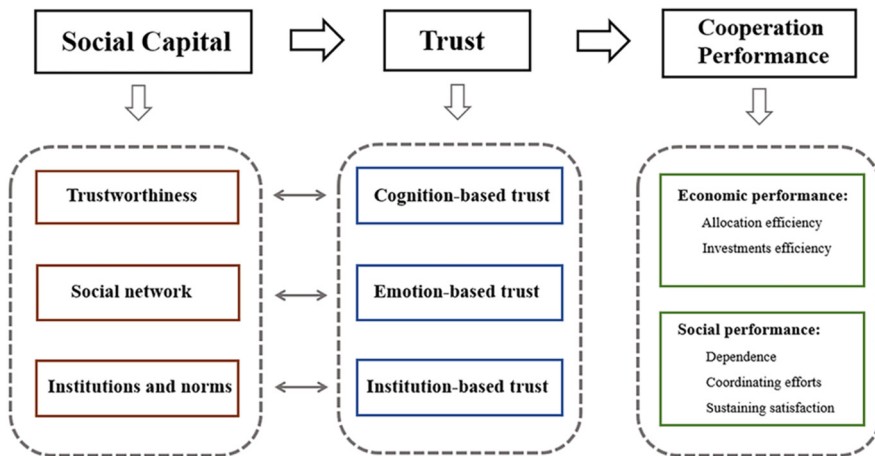

**Figure 5.** The overall conceptual or theoretical framework for the joint analysis of social capital, trust and cooperation performance.

## 5. Discussions

COCL marketization is a new attempt by the Chinese government to promote market-determined prices, self-help and orderly flows, efficient and equitable allocation of land factors, and a unified urban-rural construction land market on collectively owned rural land. The marketization narratives are such that there is a mismatch between the marketization goals and the marketization behaviour. Part of this behaviour derives from a disconnect between policy goals and collective action. This mismatch can be overcome by conceptualizing collective action in a different manner, which is specifically geared to relations of trust in a market situation.

This study on the concept and framework model addresses a critical gap in four subject areas: land administration, land sociology, agricultural sociology, and social psychology. In relation to land administration, it reveals the relationship between land and society, land systems and human behaviour from the perspective of trust, redirecting policy thinking and interventions to address collective action in land policy. This study would also contribute to land sociology and agricultural sociology, which conceptualizes social capital and acts on trust to reveal the impact of collective action on land policy. Additionally, it focuses on how trust affects cooperative performance and recognizes the important role of trust in the implementation of land policy, which enriches social psychology. The study also contributes to scientific debates on social capital, trust and cooperation performance. This joint analysis provides a full picture of the potential linkages and new knowledge in the literature on land management and land marketization.

Subjective measures of social capital, trust and cooperation performance may, to some extent, influence the effectiveness of this framework in evaluating the market-based land reform of land in China. Firstly, social capital is multifaceted and multi-disciplinary and has not yet been defined in a uniform way, with the common denominator being that it exists in the form of intangible resources in social relationships. Secondly, trust is an evolving concept, with different connotations and types in different eras, and recent scholarly definitions of trust have shifted from a focus on intentions and motivations to a focus on behaviour [70]. Finally, there are also many aspects of collaborative performance, the evaluation of which can be trapped in choosing the most efficient or the most optimal.

We examine collective action in land marketization and find that social rationality plays an important but neglected role in land marketization and that the pursuit of maximizing individual interests along the path of maximizing collective interests should be promoted in collective action. Enhancing trust can be a way to build social rationality within collective action to facilitate better government regulation of economic activity. In addition, we find that the shift from focusing on the security of land property rights to building a relationship of trust between villagers and the government is a new way of thinking to break the

collective action dilemma and innovatively propose a 'social capital-trust-cooperation performance' theoretical framework for dissecting collective action in land marketization and refining the theory of land marketization.

This study contributes to a new theoretical framework of trust-based collective action, with its underlying concepts applicable to countries amid collective action dilemmas in the implementation of land policies, especially in developing countries such as China, where land tenure systems are not yet well developed. It emphasizes the important role of social rationality and the importance of building trust between villagers and the government in the process of land marketization to promote better land economic activities by the government.

Despite the above findings, there are also possible limitations to these insights. This study only constructs a theoretical framework for collective action in rural land marketization and lacks evidence from empirical studies to support the theoretical framework of social capital-trust-cooperation performance. The next step of this research is therefore to use primary cross-sectional data from the pilot areas to answer what the roles of social capital in building mutual trust between policy implementers and landowners are and how far trust needs to go before leading to cooperation performance. Specifically, the empirical analysis of the impact of social capital on trust, in addition to answering how trustworthiness, internal and external networks, and the awareness of institutions and norms affect trust, also seeks to explore whether reciprocity may simply exist if there is no trust. Another part of the empirical research on the relationship between trust and collaboration performance focuses on the impact of different types of trust on collaboration performance and its extent.

## 6. Conclusions

We posit that the adapted conceptual framework for the joint analysis of social capital, trust and cooperation performance would be appropriate to evaluate the market-based land reform in China. This framework detects three levels of understanding of the collective action of land marketization-social capital, trust and cooperation performance, and elaborates how these relate. First, we parse the concepts of social capital, trust and cooperation performance and further classify them. It is found that social capital contains three elements: structural, cognitive and relational, trust consists of three types: emotion-based, institution-based and cognition-based and cooperative performance has two forms: economic performance and social performance. Secondly, the framework describes the connection between social capital and trust and extends current insights by making an analogy between the three elements of social capital and the three types of trust. Finally, the overall conceptual and theoretical framework for the joint analysis of social capital, trust and cooperation performance for collective action is constructed for land marketization. It is worth noting that while the framework constructed in this paper uses the example of the market-based land reform in China, it is equally applicable to land reforms associated with collective action in countries where land has been privatized. This is because COCL marketization in China is designed to activate rural land in the transaction process, while western capitalist countries, represented by Britain and the United States, have carried out land privatization reforms though, also to facilitate the capitalized flow of the land. In addition, some Commonwealth of Independent States (CIS) and Central and Eastern European socialist countries have a more similar background to China in that they generally suffer from imperfect land markets and unclear property rights, which exacerbate the dualistic structure of agricultural land [71] and the lack of clear and transferable property rights [72]. The view of the role of trust in collective action, highlighted in this paper, is somewhat free from the constraints of unclear property rights.

Obviously, we also acknowledge the limitation due to chosen methodology and literature repositories. Although we attempted to aggregate research findings from different literature repositories in different fields, this bias was not spared in the screening and review conducted by individuals. Furthermore, due to the large volume of initial literature

screened, we were unable to ensure that all worthwhile literature was included in either the title screening or the keyword and abstract screening.

This study, which is only at a theoretical and literature review level, is the beginning of a new research agenda. The next step is to rely on this theoretical framework and select suitable cases for field research with a view to analyzing the relationship between social capital, trust, and cooperation performance through empirical analysis. Considering that the elements of social capital, types of trust and forms of cooperative performance involved in collective action in land marketization are proposed only at a conceptual level, without quantifying them, future work will develop indicators to measure them. Specifically, cases of COCL marketization can be selected to obtain first-hand information through fieldwork using parallel mixed-methods techniques such as saturation logic, triangulation logic, observations, and statistics. The empirical research following the data collection is discussed in two parts. The first part of the empirical research aims to explore the influence of three different dimensions of social capital on building mutual trust between policy implementers and landowners. The second part is a dissection of the relationship between trust and cooperation performance in the land marketization process, with data on trust as described earlier and data on cooperation performance including economic performance and social performance. We not only measure cooperation performance to dissect the outcomes of collective action but also explore where/how far trust needs to go before it has a positive impact on collaboration performance.

**Author Contributions:** This manuscript is part of ongoing Ph.D. research. The candidate L.Z. and the supervisor W.T.d.V. made their respective contributions to the manuscript as follows: Conceptualization, L.Z.; methodology, L.Z.; validation, W.T.d.V.; formal analysis, L.Z.; investigation, L.Z.; resources, L.Z. and W.T.d.V.; data curation, L.Z.; writing—original draft preparation, L.Z.; writing—review and editing, W.T.d.V.; visualization, L.Z. and W.T.d.V.; supervision, W.T.d.V. All authors have read and agreed to the published version of the manuscript.

**Funding:** TUM Open Access provided funds for the publication (TUZEBIB). There was no additional funder for executing the research.

**Institutional Review Board Statement:** Not applicable.

**Informed Consent Statement:** Not applicable.

**Data Availability Statement:** Not applicable.

**Acknowledgments:** This study was carried out while undertaking a Ph.D. research program at the Chair of Land Management, Technical University of Munich (TUM). We wish to express our appreciation to the China Scholarship Council (No. 201908440280) for funding doctoral studies. We would also like to thank various experts, mentors, fellow doctoral candidates, and reviewers whose intellectual discussions and constructive comments helped to improve the paper.

**Conflicts of Interest:** The authors declare no conflict of interest.

## Appendix A

**Table A1.** Correlation table between social capital and trust (Until 17 May).

| | Terms to the Query Preview | Web of Science Records |
|---|---|---|
| Limited to the two terms "land" and "market" | "social capital" AND "land" AND "market" | 90 |
| | "trust" AND "land" AND "market" | 291 |
| | "cooperation performance" AND "land" AND "market" | 0 |
| | "social capital" AND "trust" AND "land" AND "market" | 14 |
| | "collective action" AND "land" AND "market" | 71 |

**Table A1.** *Cont.*

| | Terms to the Query Preview | Web of Science Records |
|---|---|---|
| Limited to the term "land" | "social capital" AND "trust" AND "land" | 106 |
| | "social capital" AND "land" | 689 |
| | "trust" AND "land" | 8519 |
| | "cooperation performance" AND "land" | 0 |
| | "collective action" AND "land" | 789 |
| | "rationality" AND "land" | 495 |
| Not limited to the term "land" | "social capital" AND "trust" | 4353 |
| | "trust" AND "cooperation performance" | 10 |
| | "social capital" AND "cooperation performance" | 1 |
| | "cooperation performance" | 97 |

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
