# Peer review of "Collective Action for the Market-Based Reform of Land Element in China: The Role of Trust"

_land, doi:10.3390/land11060926_

Round 1

Reviewer 1 Report

The marketization of collectively operated construction land is a great attempt in China. This paper studies the collective action of COCL from the perspective of trust, which has certain theoretical significance. But there are still several issues to consider: 1. The data in lines 71-73 of the article is too old, so it is recommended to update or replace it with recent data; 2. The title number of this article gives people a sense of confusion. Would it be better to change the four level title number to (1) and (2)? 3. The theory related to collective action should be further elaborated in the chapeau of " collection action of CoCl marketing in China ", and the collective action in this article should be clarified. In other words, will it make the article more readable to adjust the contents of the later part to the previous part? 4. In section 4.1.3, it is not enough to explain only what are economic performance and social performance. It is necessary to further deepen their logical relationship with collective action of land marketization in combination with relevant literature.

Author Response

Thank you very much for your valuable comments. Please kindly see the attachment.

Reviewer 2 Report

Overview and general impressions

The topic is original, immediate, and with potential interest to readers, and especially policymakers and different professionals involved in the market-based reform of land. The research design is appropriate and the methods used are adequately described. The results are clearly presented. The discussion supports the results and poses issues for further research.

Discussion

I would suggest that the following additional issues might be addressed:

The paper needs a introduction to the context beyond the problems stated in the title and mentioned briefly in the text – the topic of effective land use management in the light of sustainability should be highlighted at the very beginning of the introduction and maybe a connection to the SDGs and the global importance of the topic.

It seems that it is essential to support with a more clear illustration of the conceptualization of the research, or in other words – the synthesized diagram of the model developed thus presenting how the Structural dimension of social capital, the cognitive dimension, and trust and collective action are inter-related.

The text from line 503 onwards seems more suitable for Section 3. Materials and Methods. It is also essential to mention in this section what software has been used and whether the relations outlined in the dimensions are extracted or whether a hypothesis was supported by the literature review. Maybe the paper will benefit from clearly stated research question (s) stated in this section.

Finally, but not least important, it would be reasonable to review the conclusion and go beyond the repetition of texts from the summary (mentioned in the results section and in the abstract). The final message here is vague.

Author Response

Response to Reviewer 2 Comments

Point 1: The paper needs a introduction to the context beyond the problems stated in the title and mentioned briefly in the text – the topic of effective land use management in the light of sustainability should be highlighted at the very beginning of the introduction and maybe a connection to the SDGs and the global importance of the topic.

Response 1: Thank you very much for your valuable comment. We have added several sentences at the beginning of the introduction to address the context of the study, as follows.

“Sustainable development has long been an important global issue. In 2015, the United Nations launched the 2030 Agenda for Sustainable Development, with 17 global sustainable development goals (SDGs). Land use and distribution have an impact on the environment, making the topic of effective land use management in the light of sustainability particularly important.”

Point 2: It seems that it is essential to support with a more clear illustration of the conceptualization of the research, or in other words – the synthesized diagram of the model developed thus presenting how the Structural dimension of social capital, the cognitive dimension, and trust and collective action are inter-related.

Response 2: Thank you for your suggestion. In order to clarify the issues raised in your comments, we have added and revised the elaboration of the conceptual or theoretical framework (Section 4.3):

“The trustworthiness of policy implementers, social networks, institutions, and norms together constitute the types of social capital, representing the relational, structural and cognitive dimensions of social capital respectively, which influence villagers' perception of social capital and choices of land marketization behavior. Trustworthiness of policy implementers, close social networks, and proper perceptions of institutions and norms will result in villagers’ land marketization strategies, in particular, whether or not to opt for trust. Good trust relationships motivate villagers as landowners to cooperate with policy implementers and land tenure holders in land transactions, and cooperative performance is a criterion for evaluating the outcomes of villagers' land-marketing behavior. ”

Point 3: The text from line 503 onwards seems more suitable for Section 3. Materials and Methods. It is also essential to mention in this section what software has been used and whether the relations outlined in the dimensions are extracted or whether a hypothesis was supported by the literature review. Maybe the paper will benefit from clearly stated research question (s) stated in this section.

Response 3: Thank you very much for your valuable comments. We have added some contents in Section 3 to clearly state the material and method. First, I put the text from line 503 in Section 3.1. Second, we also have mentioned the software used in Section 3.3 by adding ”Literature was first removed on topics not relevant to this study by using Citavi during the importation of citation text files”. Third, we have added one sentence at the end of Section 3.3 to explain how the relations outlined in the dimensions are obtained, as follows:

‘’It is worth noting that the categories of social capital and trust are extracted separately in this process, and access to the concepts of the different categories facilitates an in-depth dissection of the connection of social capital and trust.‘’

Point 4: Finally, but not least important, it would be reasonable to review the conclusion and go beyond the repetition of texts from the summary (mentioned in the results section and in the abstract). The final message here is vague.

Response 4: Thank you for pointing out the issue. We replaced the description of the data in the final message of previous versions with the outlook for two empirical studies, helping readers to follow up on our subsequent research, as follows.

‘’The empirical research following the data collection is discussed in two parts. The first part of the empirical research aims to explore the influence of three different dimensions of social capital on building mutual trust between policy implementers and landowners. The second part is a dissection of the relationship between trust and cooperation performance in the land marketization process, with data on trust as described earlier and data on cooperation performance including economic performance and social performance. We not only measure cooperation performance to dissect the outcomes of collective action but also explore where/how far trust needs to go before it has a positive impact on collaboration performance. ‘’

Reviewer 3 Report

It is a really very interesting topic. The weakness of the article is its explicit theoretical approach. Some general data on the main features of expropiation process (cases, surfaces, population affected...) could help to understand the real challenge of the Government policy. From theoretical point of view, as authors aknowledge theyselves never could be exhaustive; every reader coul find other sources and interpretations, Finally, some review of the same process in other countries (historically in western countries or more recently in former socialist countries) could enlarge the conclusions.

Author Response

Response to Reviewer 3 Comments

Point 1: It is a really very interesting topic. The weakness of the article is its explicit theoretical approach. Some general data on the main features of expropiation process (cases, surfaces, population affected...) could help to understand the real challenge of the Government policy. From theoretical point of view, as authors aknowledge theyselves never could be exhaustive; every reader coul find other sources and interpretations, Finally, some review of the same process in other countries (historically in western countries or more recently in former socialist countries) could enlarge the conclusions.

Response 1: Thank you for your understanding that the theoretical approach never could be exhaustive. We did some review of the same process in other countries (historically in western countries or more recently in former socialist countries) to enlarge the conclusions in Section 6, as follows:

“It is worth noting that while the framework constructed in this paper uses the example of the market-based land reform in China, it is equally applicable to land reforms associated with collective action in countries where land has been privatized. This is because COCL marketization in China is designed to activate rural land in the transaction process, while western capitalist countries, represented by Britain and the United States, have carried out land privatization reforms though, also to facilitate the capitalized flow of the land. In addition, some Commonwealth of Independent States (CIS) and Central and Eastern European socialist countries have a more similar background to China in that they generally suffer from imperfect land markets and unclear property rights, which exacerbate the dualistic structure of agricultural land[71]and the lack of clear and transferable property rights[72]. The view of the role of trust in collective action, highlighted in this paper, is somewhat free from the constraints of unclear property rights.”
